# The biological function of an insect antifreeze protein simulated by molecular dynamics

Michael J Kuiper[1]*, Craig J Morton[2], Sneha E Abraham[3], Angus Gray-Weale[3]

[1]Victorian Life Sciences Computation Initiative, The University of Melbourne, Carlton, Australia; [2]ACRF Rational Drug Discovery Centre, St Vincent's Institute of Medical Research, Fitzroy, Australia; [3]School of Chemistry, The University of Melbourne, Melbourne, Australia

**Abstract** Antifreeze proteins (AFPs) protect certain cold-adapted organisms from freezing to death by selectively adsorbing to internal ice crystals and inhibiting ice propagation. The molecular details of AFP adsorption-inhibition is uncertain but is proposed to involve the Gibbs–Thomson effect. Here we show by using unbiased molecular dynamics simulations a protein structure-function mechanism for the spruce budworm *Choristoneura fumiferana* AFP, including stereo-specific binding and consequential melting and freezing inhibition. The protein binds indirectly to the prism ice face through a linear array of ordered water molecules that are structurally distinct from the ice. Mutation of the ice binding surface disrupts water-ordering and abolishes activity. The adsorption is virtually irreversible, and we confirm the ice growth inhibition is consistent with the Gibbs–Thomson law.

## Introduction

Many species of fish, insects, plants and micro-organisms living in cold environments produce antifreeze proteins (AFPs) whose main function is to target and modify the growth of regular ice (*Raymond and DeVries, 1977*; *Davies and Hew, 1990*). By binding to ice, AFPs are thought to both lower the freezing point and raise the melting point of ice through the Gibbs–Thomson effect whereby adsorbed AFPs cause increased micro-curvature of the ice front (*Yeh and Feeney, 1996*). The resulting difference between the melting and freezing points, known as the thermal hysteresis gap (TH), can range from a few tenths to over 6°, depending on the specific AFP (*Lin et al., 2011*).

Regular hexagonal ice ($I_h$) has three equivalent *a*-axes ($a_1$–$a_3$) which are parallel to three respective prism planes and perpendicular to the *c*-axis. The ice basal plane lies perpendicular to the *c*-axis. Remarkably AFPs have evolved independently numerous times to bind stereo-selectively to ice in various orientations, resulting in a structurally diverse protein family (*Hew et al., 1985*; *Ko et al., 2003*; *Pentelute et al., 2008*; *Hakim et al., 2013*; *Middleton et al., 2013*). The potency of different AFPs reflects their adsorption characteristics and environmental adaptation. Polar fish, for example, do not normally experience temperatures much lower than the freezing point of sea water (−1.9°C), and have AFPs that provide protection to just a few tenths of a degree below this. Fish AFPs often bind oblique to ice prism planes, typically forming hexagonal bipyramid crystals at temperatures within the TH gap which rapidly grow as elongated spicules along the *c*-axis below the TH gap (*Davies and Hew, 1990*). In contrast insects which are exposed to much lower terrestrial temperatures, (often below −30°C), require AFPs with higher activity, possibly by selecting more effective prism and basal plane binding (*Lin et al., 2011*). Despite a wide selection of available structural and experimental data, a complete molecular description of how an AFP binds to its respective ice planes and inhibits crystal growth remains elusive.

*For correspondence: mkuiper@unimelb.edu.au

Competing interests: The authors declare that no competing interests exist.

**eLife digest** Water expands as it freezes. If this happens to the water inside plants and animals, the resulting ice crystals can rupture cells. To prevent this, many plants and animals that live in cold climates have evolved 'antifreeze proteins'. When a small particle of ice first starts to form, the antifreeze proteins bind to it and prevent the water around it freezing, hence preventing the growth of an ice crystal.

There are many different types of antifreeze protein, and some are more active than others. For example, some insects including the spruce budworm are exposed to extremely cold temperatures—sometimes below −30°C—and these insects have antifreeze proteins that are highly active.

It is not fully understood how different antifreeze proteins interact with ice and prevent the growth of ice crystals. This is largely because, as yet, there are no experimental techniques that make it possible to see how antifreeze proteins and water molecules arrange themselves at the surface of a growing particle of ice. Instead, scientists have developed computer simulations to investigate this process. While many of these studies have provided valuable information, the computational methods used have only recently become powerful enough to analyze how the antifreeze proteins approach the surface of the ice particle.

Kuiper et al. carried out simulations involving a highly active antifreeze protein from the spruce budworm. The results of these simulations revealed that this antifreeze protein does not bind directly to ice; instead, water molecules at the surface of the protein act as a bridge between the protein and the ice. These water molecules are highly ordered and though they have similarities with how water is structured in the ice, they are distinct from the ice lattice itself. Furthermore, this arrangement appears to be important for allowing the spruce budworm antifreeze protein to interact with the ice.

This study provides detailed insights as to how a highly active antifreeze protein helps to prevent ice crystals forming. In the future, the computational simulations used here may be extended to study the dynamics of other antifreeze proteins, and also how crystals of other materials form.

The difficulty in characterizing the molecular mechanisms of AFPs arises largely due to their unusual relationship with water, which in this case serves as both the protein's solvent and, as ice, its target (*Jia and Davies, 2002*). Slowly freezing solutions of dilute AFP will incorporate AFPs into the growing ice phase at roughly their original concentration, while non-AFPs and solutes are largely excluded (*Marshall et al., 2004*). Many structural studies have shown a correlation between the periodicity of the ice-binding surface elements of AFPs and their respective ice-binding planes, though it has not been entirely clear whether AFPs bind directly to ice or through some ordered water intermediate (*Sharp, 2011*). Observation of the molecular arrangements of AFPs and surrounding water as they bind and modify the surface curvature of ice is currently technically unattainable. Given these experimental limitations, computer simulations have played a crucial role in expanding our understanding of the AFP phenomena.

Early computational simulations of winter-flounder AFP (wfAFP) by *Wen and Laursen (1992)*; *Jorgensen et al. (1993)*; *Madura et al. (1994)* firmly established the correlation of the distance between regularly spaced polar threonine residues and that of water molecules of the ice lattice of the {2021} pyramidal ice plane in the [112] direction by searching for energetically stable conformations. This was in agreement with previous experimental work of Knight et al which showed the preferential binding orientation of wfAFP on single crystal ice hemisphere (*Knight et al., 1991*) and also the later work of Sicheri and Yang who determined the wfAFP crystal structure (*Sicheri and Yang, 1995*). Cheng and Merz further expanded this work using molecular dynamics simulations of solvated wfAFP in the bound conformation to the {2021} pyramidal plane to propose key hydrogen bond interactions between polar residues and the ice surface, suggesting that hydrogen bonds were the primary driving force of ice adsorption (*Cheng and Merz, 1997*). However, this model was to be short lived as mutational work by the groups of Harding and Laursen (*Zhang and Laursen, 1998*; *Haymet et al., 1999*) raised questions about the nature of the wfAFP-ice interaction; conservative substitution of threonine residues for serine abolished activity, while threonine to valine substitution, which removed the proposed threonine hydroxyl–ice interactions surprisingly retained AFP activity.

Spurred by the paradoxical mutant wfAFP results both *Dalal and Sönnichsen (2000)* and *Jorov et al. (2004)* employed Monte Carlo simulations of the wfAFP on the pyramidal plane, both groups finding that hydrophobic interactions contributed significantly to the adsorption to ice. Further related work by *Wierzbicki et al. (2007)* proposed that wfAFP doesn't bind directly to ice but rather accumulates at the ice-water interface with preference for the hydrophobic over the hydrophilic face facing the ice. Additional computational studies of different AFPs have followed, including type II sea-raven AFP (*Wierzbicki et al., 1997*), type III eel pout AFP (*Madura et al., 1996*) and an insect AFPs from *Tenebrio molitor* (*Liu et al., 2005*). Nada and Furukawa (*Nada and Furukawa, 2011*) were amongst the first to provide a model of the spruce budworm AFP (sbwAFP) at the prism ice-water interface. Using rigid models of sbwAFP they showed that two pre-determined binding conformations were able to affect local ice growth kinetics, inferring that reduced ice growth and induced curvature was indicative freezing point depression consistent of the Gibbs–Thomson effect.

As highlighted by an excellent review by Nada and Furukawa. (*Nada and Furukawa, 2012*), in real world systems, AFPs bind to an ice-water interface rather than ice crystal planes alone, thus earlier AFP simulations are unable to provide complete mechanistic details of the AFP-ice interaction. Even though a number of AFP studies have simulated ice-water interactions (*McDonald et al., 1995*; *Dalal et al., 2001*; *Wierzbicki et al., 2007*; *Nada and Furukawa, 2008*), Nada and Furukawa also point out the need for simulations to target a growing ice-water interface. Ideally, simulations would allow unconstrained AFPs to migrate from the water phase to interact freely the growing ice-water interface without introducing prior binding orientation bias. The computational requirements to do so are significant, and only recently became relatively accessible for researchers.

The highly active AFP from the spruce budworm, *Choristoneura fumiferana*, (sbwAFP) provides an excellent model for antifreeze simulations and has been used previously in computational studies (*Nutt and Smith, 2008*; *Nada and Furukawa, 2011*). SbwAFP has a β-helix structure with a triangular cross-section forming three parallel β-sheet faces and a hydrophobic core as shown in *Figure 1*. The ice binding face has a regular array of repeating threonine-x-threonine (T-X-T) motifs whereby the spacing between threonine residues match the repeat spacing parallel to the *c*-axis of ice of approximately 7.4 Å and motifs on adjacent β-strands closely match the *a*-axis repeating dimensions of the prism face of ice of 4.5 Å. The second row of threonine residues of the ice-binding surface is particularly sensitive to mutation, with replacement of the threonines drastically reducing sbwAFP's potency as an AFP (*Graether et al., 2000*). In addition to inducing a non-colligative freezing point depression of ice by up to 6°C, sbwAFP also demonstrates a significant melting inhibition (super-heating) of ice of up to 0.44°C (*Celik et al., 2010*). Here we employ multiple and extended molecular dynamics simulations of a potent sbwAFP isoform 501 (*Leinala et al., 2002*) with a novel ice/water interface model designed to create continual ice growth conditions, providing new insight into molecular specificity and the resultant Gibbs–Thomson induced thermal hysteresis.

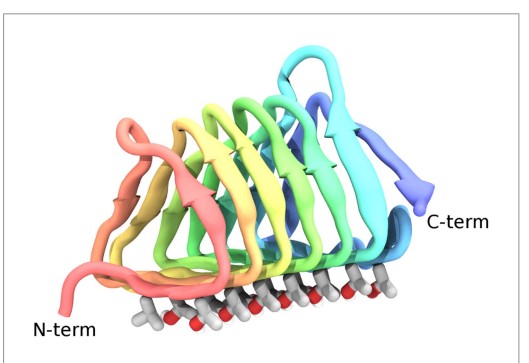

**Figure 1**. Cartoon representations of the spruce budworm AFP (sbwAFP) 501 isoform. This shows the triangular cross section of parallel β-helix structure. The ice binding surface is drawn as sticks.

## Results and discussion

### Building a model of the ice-water interface

The TIP4P water model (*Jorgensen et al., 1983*) was used for simulations due to its ability to give realistic phase transformation behavior and superior freezing characteristics compared with the default TIP3P model, its support by the NAMD software (*Phillips et al., 2005*), and its compatibility with biological CHARMM parameters (*Glass et al., 2010*). Although the melting point ($T_m$) of the TIP4P water model is known to be 230.5 K, (*García Fernández et al., 2006*) it is highly unlikely such a water model will spontaneously freeze or remain frozen in simulation, particularly in the few degrees temperature range below $T_m$ where AFPs exhibit their biological activity, due to the equilibrium radius $R$ of the critical ice nucleus

governed by the Gibbs–Thomson effect. This effect is a well known phenomenon whereby the chemical potential across an interface varies with curvature. For positive interfacial energies smaller crystals, with their higher curvature, are only able to be in equilibrium with their melt at lower temperatures than larger crystals. AFPs are thought to exploit this property by effectively subdividing a large ice surface of relatively low curvature into smaller surfaces with greater curvature thus giving rise to lower freezing temperatures, as shown in *Figure 2*. The critical radius at a given temperature $T_r$ can be approximated with the following expression adapted from *Pereyra et al. (2011)*:

$$R \approx \frac{M_w \sigma}{L \rho_i} \frac{A_g T_0}{(T_0 - T_r)}, \tag{1}$$

where $M_w$ is the molecular weight of water, $\sigma$ is the surface tension of the ice-water interface, $L$ is the latent heat of fusion, $\rho_i$ is the density of ice, $T_0$ is the melting temperature at atmospheric pressure, $A_g$ is a constant of 2 or 1 for spherical or cylindrical symmetry conditions respectively. This relationship implies large ice crystals are required to maintain growth. For example, assuming spherical geometry, for a single degree of supercooling the critical radius of an ice embryo is about 518 Å containing approximately 18 million water molecules, which is currently computationally prohibitive to model (see 'Materials and methods' for calculation). To simulate AFP-ice interactions more efficiently a specially arranged model of an ice-water interface was developed to provide a continual ice growth mechanism while mimicking an infinite ice plane by taking advantage of periodic boundary conditions (PBCs). The ice/water system employed here consisted of 17,434 water molecules in a 51.5 × 126.6 × 83.9 Å box with PBCs, of which 1692 water molecules were harmonically constrained to an ideal prism ice lattice. This served as the seed crystal while an additional 1049 water molecules below the seed were constrained in a 5 Å thick disordered layer to act as an ice barrier preventing downward ice growth, leaving a small wedge of mobile water between constrained layers as shown in *Figure 3*. A perspective view of the model is shown in *Figure 3—figure supplement 1*. The seed ice crystal included an additional small prism step to initiate step layer growth and the entire seed was angled such that the primary prism face was approximately 4.3° with respect to the long axis. The angle of the seed is such that step layer growth across the prism face reappears, due to PBCs, one layer higher to where it started, thus providing a continual ice growth mechanism. Unimpeded, this mechanism will freeze the area above the seed crystal at any temperature below $T_m$. This arrangement mimics step growth as might be produced by screw dislocations, as originally proposed by *Frank (1949)* and later observed on ice surfaces by *Ketcham and Hobbs (1968)*.

As the simulation proceeds, water molecules join the ice phase through hydrogen bonding and can be easily identified by their tetrahedral arrangement, which appears hexagonal when viewed directly down the *c*-axis of ice. Within this ice/water box the unconstrained sbwAFP model was initially placed 20 Å above the seed, with its ice binding face directed towards the seed approximately aligned in the expected binding orientation. In our preliminary work the seed ice crystal was arranged with the prism face parallel with the axis of the simulation boundaries. Crystals from these arrangements did not grow well, extending the ice front very slowly if at all. We realized that this was because one had to effectively wait for a two-dimensional nucleation event to occur on the prism plane and reach a critical size before the next layer of ice would grow.

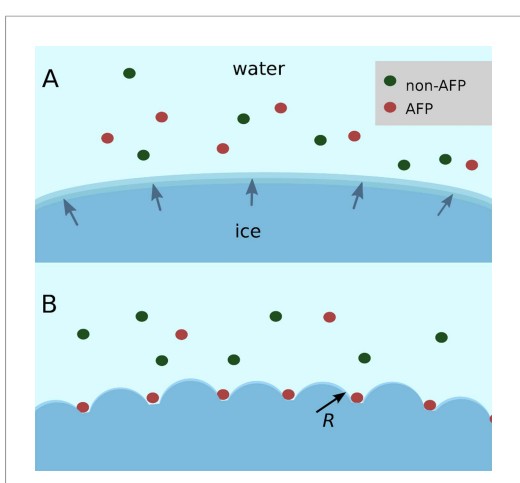

**Figure 2**. Schematic figure of antifreeze protein (AFP) ice inhibition via the Gibbs–Thomson effect. (**A**) A mixture of AFP and non-AFP proteins ahead of a growing ice front. (**B**) AFPs selectively bind to the ice, while non-AFPs are excluded. AFP adsorption to the ice surface subdivides into smaller growth fronts which increases local curvature. If the induced curvature of the new ice fronts is less than the critical radius (*R*) of an ice embryo at a given temperature then the ice remains in equilibrium with the melt.

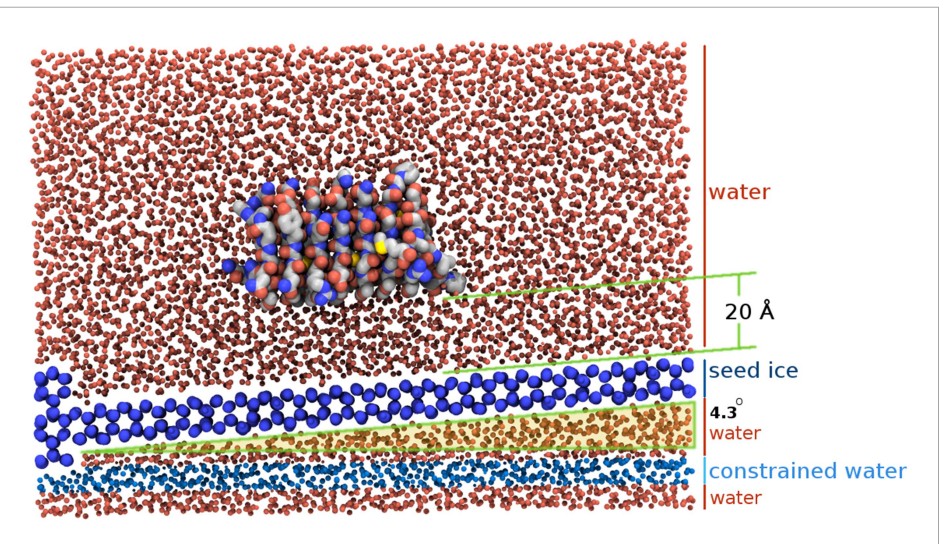

**Figure 3**. Section view of the sbwAFP simulation setup. The seed ice crystal is shown in dark blue, constrained disordered water coloured cyan and unrestrained water is coloured red. The sbwAFP is centrally positioned 20 Å above the seed ice. The prism face of the seed ice crystals is angled approximately 4.3° with respect to the long axis of the simulation box, such that the periodic image of the seed ice is consistent with itself at the edges.

The following figure supplement is available for figure 3:

**Figure supplement 1**. Perspective view of the model showing the seed ice as dark blue spheres and free waters as transparent surface.

## Characterization of the sbwAFP binding mode

To characterize the binding mode of sbwAFP, 32 independent, unconstrained (apart from the waters constrained in the ice seed and ice-barrier) simulations of at least 150 nanoseconds (ns) to as long as 250 ns duration were carried out at 228 K. Definitive ice binding was observed 26 times with each binding event occurring with equivalent orientation with respect to the ice lattice, though adsorbing at a number of different step heights. The remaining 6 simulations either diffused away from the growing ice front (3 times), or had partial engagement at alternate angles (3 times). Of the partial engagements, the AFP still had significant rotational and translational movement with respect to the ice lattice so was not considered truly bound. Each of the definitive ice binding events incorporated a linear array of 6 water molecules bridging between the prism face and a row of conserved threonine residues (Thr 7, 23, 39, 54, 69, 84 and 101) as shown in *Figure 4*. These water molecules had the same periodicity of the prism face, spaced approximately 4.5 Å apart. Interestingly they were distinct from the bulk ice crystal, bridging between two prism face water molecules parallel along the *c*-axis which would normally be bridged by two water molecules in regular ice as shown in *Figure 5*. The sbwAFP-bound water molecules had fully satisfied tetrahedral hydrogen bonding arrangements; two hydrogen bonds to the prism face ice water molecules and two hydrogen bonds to the hydroxyl groups of adjacent threonine residues of the sbwAFP ice binding face.

To further investigate the function of the ordered waters of the ice binding surface 16 independent simulations of a sbwAFP mutant (with 4 Thr to Leu substitutions at positions 7, 21, 39, 69) known to disrupt activity (*Graether et al., 2000*), were performed, each of 250 ns length. No ice binding events were observed with all mutant AFPs diffusing ahead of the growing ice surface in distinct contrast to the wild type protein. *Figure 6* shows the relative height coordinates of both wild type and mutant sbwAFP throughout the docking simulations clearly demonstrating the former AFP adsorbing and fixing to the ice surface while the latter, unable to bind, diffuses ahead of the growing ice front. The Thr to Leu substitutions disrupt the binding of the ordered water to sbwAFP by removing the stabilizing hydroxyl side chain hydrogen bond interactions of the threonine residues. These results indicate ordered waters are crucial in initiating sbwAFP recognition and binding to ice and are

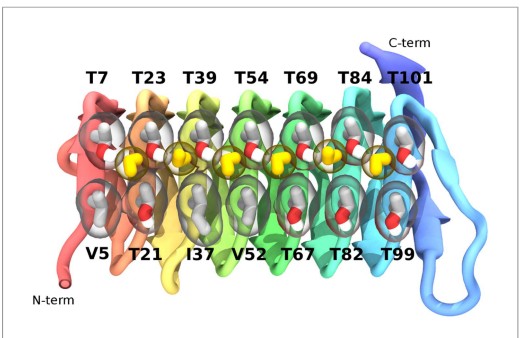

**Figure 4.** The ice-binding surface of sbwAFP. Ordered water molecules are shown in yellow hydrogen bonded between adjacent threonine residues of the top row.

suggestive of longer range protein-water interactions consistent with recent experimental findings by Meinster et al, of a similar beetle AFP from *Dendroides canadensis*, measured with terahertz spectroscopy (*Meister et al., 2013*). Representative simulations of the binding and non-binding events of the respective wild type and mutant sbwAFP are shown in supplementary *Videos 1, 2*.

## Adsorption inhibition

Three contiguous ~1 µs MD simulations of sbwAFP adsorbed to an active ice front at temperatures of 225 K, 230 K and 232 K were run to test and simulate the adsorption-inhibition mechanism by observing subsequent ice growth as shown in *Figure 7*. Representative cross-section snapshots of the simulation at the three temperatures is shown in *Figure 7—figure supplement 1*. Though the Gibbs–Thomson phenomena is widely accepted as the basis of AFP activity, to the best of our knowledge it has not been observed directly at the ice-water interface. In the first simulation the sbwAFP adsorbed from the solvent phase to the advancing ice front at 225 K (5 K below the model system $T_m$). The sbwAFP adsorbed to the ice front within 40 ns and within 150 ns the ice front displayed marked convex curvature creating cylindrical ice fronts as shown in *Figure 8*. By 250 ns the ice front had reached equilibrium with an averaged cylindrical radius of approximately 50 Å. The extent of the inhibited ice front fluctuated, advancing and retreating in the range of 5–8 Å showing the reversible arrangement of water molecules to the crystal ice lattice, (this is an important indication of reaching equilibration and adequate sampling given the reduced water diffusion at these low temperatures). The ice fraction (defined as a proportion of total ice-like water molecules above the ice seed crystal), was measured over the 280 to 1000 ns time frame and found to equilibrate at 0.46 ± 0.01 (SD). In contrast AFP-free

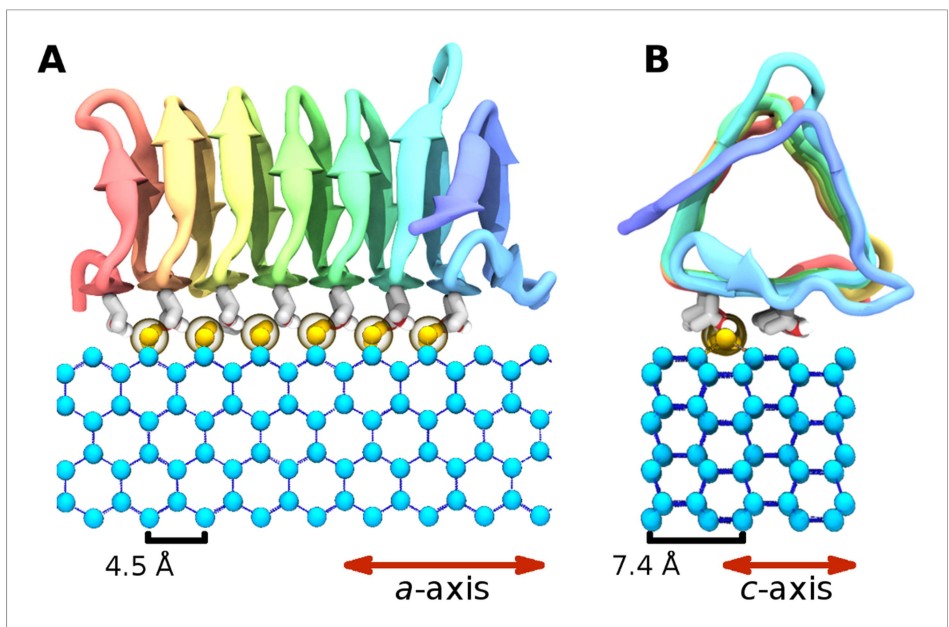

**Figure 5.** The ice binding arrangement of spruce budworm AFP in relation to ice. (**A**) *c*-axis view of ice (cyan spheres) and the arrangement of the sbwAFP ordered water (yellow spheres). (**B**) *a*-axis view of the sbwAFP binding orientation showing the relative arrangement of the ordered water to ice. The ice lattice repeats approximately 4.5 Å and 7.4 Å along the a axis and c axis respectively.

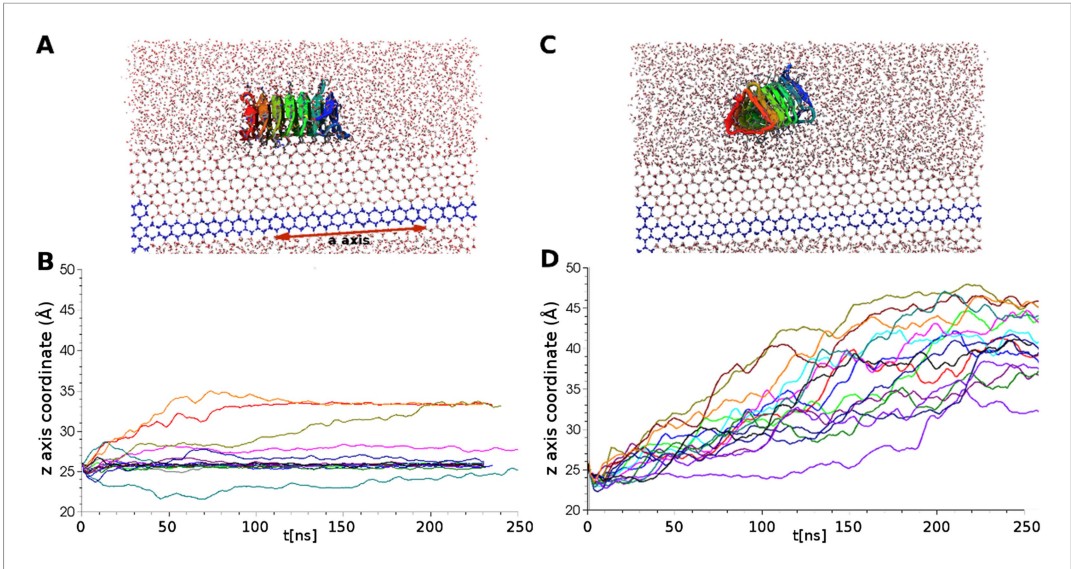

**Figure 6**. Simulations of ice docking of wildtype and inactive mutant sbwAFP against a growing ice surface. (**A**) Representative side view of final docking orientation of wild type sbwAFP. (**B**) Overlay of 16 independent wild type sbwAFP docking simulations mapping the centre of mass z-axis coordinate of the AFP over time, an unchanging z-coordinate shows ice binding. (**C**) Representative side view of mutant sbwAFP in simulation. (**D**) Overlay of 16 independent mutant sbwAFP docking simulations showing all AFPs diffusing ahead of the ice front and no ice binding events. The mutant form of the sbwAFP consisted of 4 Thr to Leu mutations at positions 7, 21, 39 and 69 which disrupts the ice binding surface.

simulations froze almost completely within 150 ns. The vacillating nature of the ice front in simulation makes quantitative measurement of the curvature difficult, however we find good qualitative agreement between theoretical Gibbs–Thomson curvature and the averaged ice front as shown in *Figure 8* where the theoretical radius is overlaid on to the averaged simulation ice cross-section. This is good indication that the simulation ice curvature is due to the expected Gibbs–Thomson phenomena.

When the temperature was raised to 230 K, (the approximate system melting point $T_m$) the simulation showed a significant reduction of the ice fraction to an equilibrated value of 0.21 ± 0.02, (calculated over the 1100 to 2130 ns time frame) shown in *Figure 8B*. The sbwAFP remained bound to ice, stabilizing an ice protrusion directly beneath the AFP approximately 7 Å high relative to the main ice front.

In the final contiguous simulation the temperature was raised to 232 K, (approximately 2 K above the simulation melting point) and showed further reduction in the ice fraction which equilibrated to an average of 0.15 ± 0.01 (calculated over 2190 to 3160 ns time frame). The sbwAFP remained attached to the ice protrusion approximately 8.5 Å above the main ice plane which developed slight concave curvature with respect to the adsorbed protein (*Figure 8C*). A continuation of the simulation at 235 K resulted in the detachment of the sbwAFP from the ice surface within 10 ns allowing further melting of the ice to a residual fraction of 0.08 ± 0.01. Steady-state representations of the equilibrated ice fronts for

**Video 1.** Representative simulation of spruce budworm AFP (sbwAFP) docking to ice. Side view of unconstrained sbwAFP docking to the prism face of a growing ice surface at 228 K. (250 nanosecond total simulation.) The seed ice crystal is in dark blue.

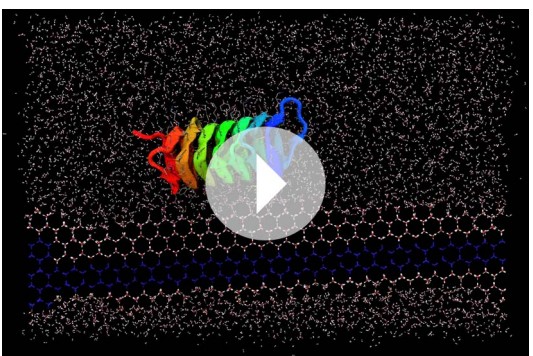

**Video 2.** Representative simulation of mutant sbwAFP. Unconstrained sbwAFP mutant simulated at 228 K for 250 nanoseconds. Ice binding ability has been lost by mutation of 4 threonine residues (7, 21, 39 and 69) of the ice binding surface to leucine.

each temperature were generated averaging water density over each simulation segment, by mapping water occupancy of greater than 85% using the 'volmap' module of the visualization software VMD (*Humphrey et al., 1996*). Extended simulations showing the induced curvature of the ice surface are shown in *Videos 3, 4, 5*. These simulations clearly demonstrate the Gibbs–Thomson effect showing how adsorbed protein induces ice-surface curvature and subsequent melting and freezing inhibition of ice. An important aspect of AFP activity that remains to be resolved involves the question of the strength of binding of AFP to ice. Previous estimations of the binding energy of AFPs to ice have varied greatly in the literature. Initial calculations from *Cheng and Merz (1997)* put the wfAFP binding energy of −157 kcal/mol while *Dalal and Sönnichsen (2000)* put the figure at −67 kcal/mol. Later estimates by *Jorov et al. (2004)* and *Wierzbicki et al. (2007)* both put the figure for wfAFP much lower at below −5 kcal/mol. Such low values of binding, however, imply an equilibrium with appreciable amounts of exchange of the AFP from the ice surface and its surroundings. Contrary to this, experimental evidence suggests that the binding energy must be relatively high and virtually irreversible. Microfluidic experiments by *Celik et al. (2013)* show that ice growth remains inhibited despite the exchange of the solution surrounding AFP-inhibited ice crystals with AFP-free buffer, suggesting there is little or no exchange of AFPs from the ice surface once bound. This finding is further strengthened by recent field observations of Cziko et al. where it was found Antarctic notethenoid fishes internally accumulated ice does not melt as expected over summer warming periods (*Cziko et al., 2014*). Our own observations in this study suggest relatively strong binding of a magnitude similar to that proposed by

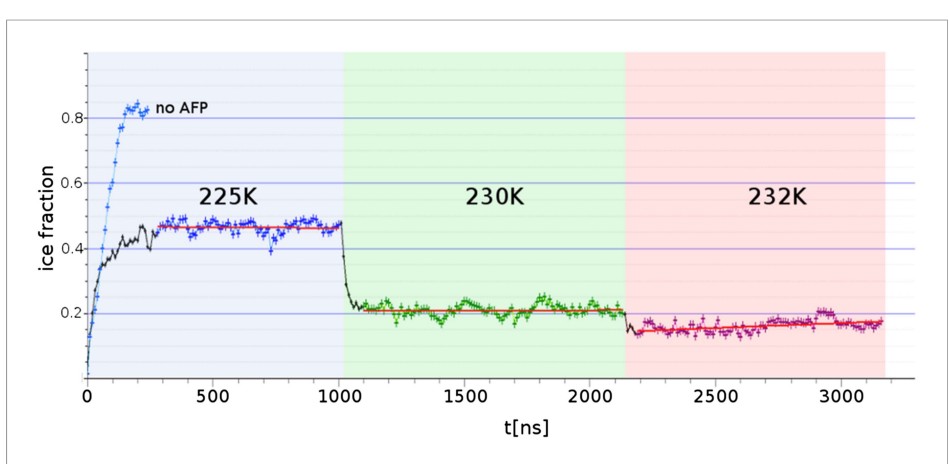

**Figure 7**. Freezing and melting inhibition of ice by adsorbed sbwAFP. Ice fraction of 3 contiguous ~1 microsecond simulation temperatures 225 K (approx. 5 K below melting point $T_m$), 230 K (approx. at $T_m$) and 232 K (approx. 2 K above $T_m$). Linear regression of equilibrated states shows no appreciable growth or melting (red lines). A simulation containing no AFP (blue line on left) shows rapid freezing of the entire model.
The following figure supplement is available for figure 7:

**Figure supplement 1**. SbwAFP simulation side-profile snapshots equilibrated at different temperatures.

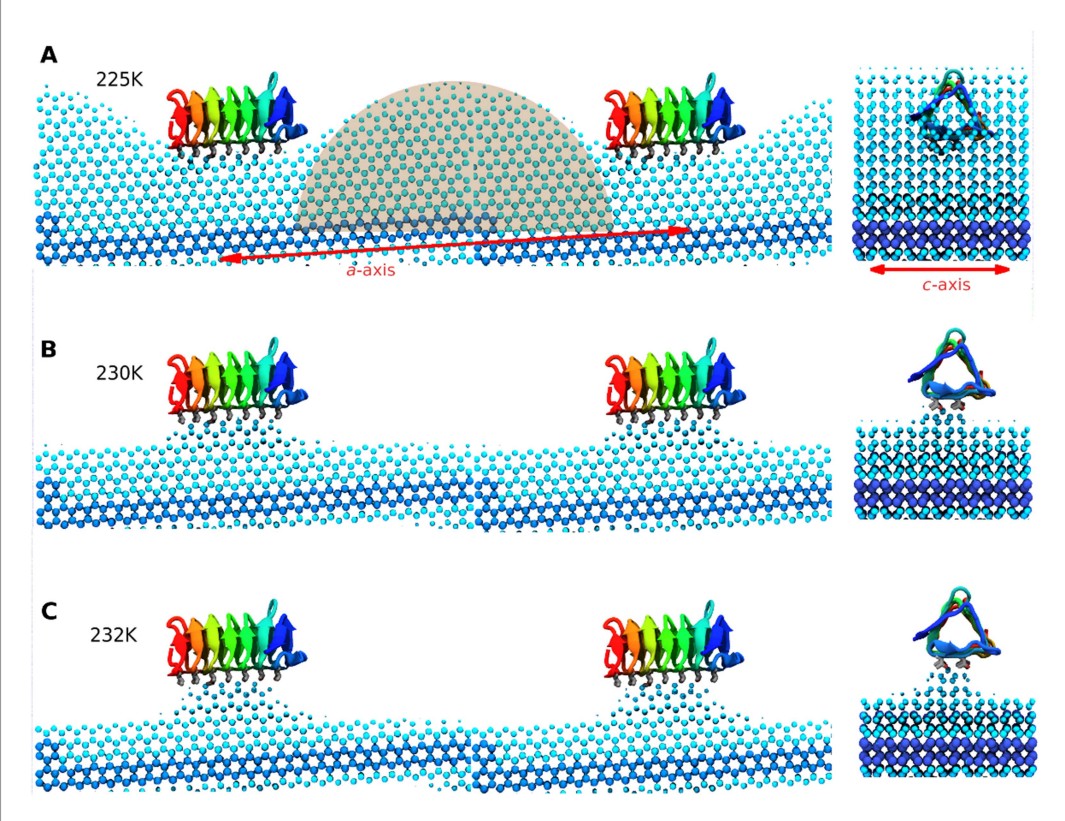

**Figure 8**. Side and end views of steady-state equilibrated ice fronts inhibited by adsorbed sbwAFP. Cyan spheres represent the averaged ice front (defined as water occupation of greater than 85% averaged over approximately 1 microsecond simulation for each temperature). Dark blue spheres represent the constrained ice seed. (**A**) At 225 K the ice front shows curvature around the adsorbed sbwAFP. The middle shaded circle segment has a radius of 46 Å equal to the theoretical Gibbs–Thomson cylindrical curvature of TIP4P water at 5 K supercooling. (**B**) At 230 K the ice retreats, but is stabilized directly beneath the bound sbwAFP. (**C**) At 232 K, (approximately 2°C above $T_m$), the bound sbwAFP remains in place stabilizing an ice protrusion, pinning back further ice retreat. The side profile views are made from two adjacent simulation periodic images.

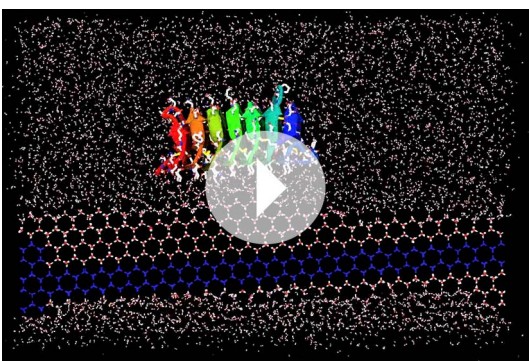

**Video 3.** Ice-bound sbwAFP simulated for 1000 nanoseconds at 232 K, showing stabilization of the ice front by the presence of the bound sbwAFP.

*Dalal and Sönnichsen (2000)*, with the sbwAFP never detaching once bound to ice, (apart from deliberate melting events), even over the course of the extended millisecond range simulations at above melting temperatures. We intend to address the magnitude of the binding energy in our future work.

## AFP activity and ordered water

Our simulations have shown that sbwAFP activity depends on binding and arranging water in a similar periodicity to ice which in turn bind to the prism face of ice, and that the ordered water plays a crucial role in determining ice plane specificity and integrating into the ice lattice. This conclusion is supported by previous work by *Nutt and Smith (2008)* whose molecular dynamics simulations

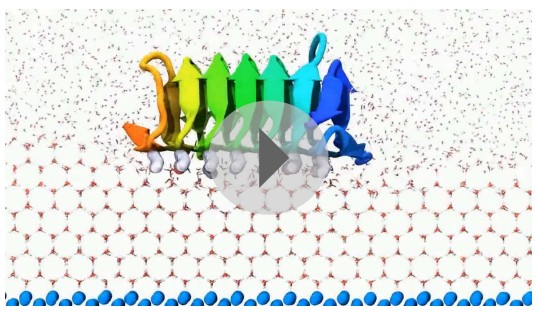

**Video 4.** Simulation of sbwAFP binding to the prism ice front at 225 K. The adsorption of the sbwAFP to the prism face causes curvature to the ice consistent with the Gibbs–Thomson effect.

of sbwAFP in solution indicated water was more ice-like on its binding face at low temperatures and speculated that this pre-configured water was important for initial recognition and binding events. Recent work by *Midya and Bandyopadhyay (2014)* simulating insect AFP from *Tenebrio molitor* with a similar ice binding surface to sbwAFP in water have also found equivalent arrangements of bound water molecules emphasizing the importance of the hydroxyl groups of the ice-binding threonine residues. Contrary to this, as mentioned previously, it is noteworthy that the putative ice-binding threonine residues of the alpha helical wfAFP can be substituted to structurally similar but hydrophobic valine residue with little effect on activity (*Haymet et al., 1999*). This shows wfAFP ice binding does not depend on the threonine hydroxyl group, even suggesting a hydrophobic mode of interaction. Though tempting to make comparisons with sbwAFP threonine residues, we should be reminded that these proteins are completely different structures with significantly different ice-binding properties. It will be important to characterize both the wfAFP/ice interaction as well as analogous sbwAFP threonine to valine mutations before valid interpretations can be made.

Many of the solved AFP crystal structures have been found to contain ordered waters which have been postulated to integrate directly into the ice lattice (*Liou et al., 2000*). In a recently determined structure for a highly active bacterial AFP from *Marinomonas primiryensis* (mpAFP), bound waters exhibited an extensive 'clathrate-like' network that matched ice lattice dimensions (*Garnham et al., 2011*). Interestingly, unlike most AFPs, mpAFP appears to bind to all ice plane orientations, rather than specific ones. This may result from extended ice-like ordering of water from the ice binding surfaces interacting with ice in non-specific orientations, perhaps in a process similar to ice sintering (*Kuroiwa, 1961*).

As noted in a recent review by *Nada and Furukawa (2012)*, previous AFP simulations have focused on modeling prearranged AFPs at the ice-water interface at equilibrium and have not yet tackled free AFP interacting with an actively growing ice interface which would reveal more mechanisms of the molecular process and be less biased to initial positioning. Our freely diffusing MD simulations of AFP spontaneously adsorbing and inhibiting an active ice surface model have demonstrated that simulations can be used to provide detailed insight into molecular processes like crystal inhibition and modification.

We have presented simulations of the sbwAFP spontaneously interacting with a growing ice surface and demonstrated key aspects of the AFP mechanisms; namely the process of molecular surface recognition and irreversible adsorption inhibition of the ice surface. We have shown that the Gibbs–Thomson formula agrees closely with the simulated inhibition of the growing ice front, and demonstrated irreversibility of binding by free energy analysis. This is one of the first computer simulations to provide a complete in silico demonstration of a protein's biological function.

## Materials and methods

### Molecular modelling

Molecular simulations were performed using NAMD2.9 (*Wierzbicki et al., 1997*) with CHARM22 (*MacKerell et al., 1998*) forcefield employing a TIP4P water model supported on

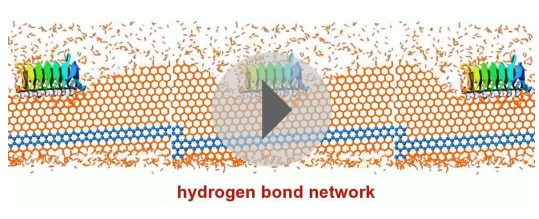

hydrogen bond network

**Video 5.** Continuation of *Video 4* showing the ice curvature with periodic boundary conditions applied. The second half of the video highlights the positions of the bound water molecules and shows how they migrate to the ice binding surface from the beginning of the simulation.

both IBM BlueGene/P and BlueGene/Q architecture. The sbwAFP model was based on the pdb structure 1M8N (*Leinala et al., 2002*). Simulations were run with PBCs using the NPT ensemble at various temperatures (225, 228, 230, 232 and 235 K) and 1 bar pressure employing Langevin dynamics. The PBCs were constant in the XY dimensions. Long-range Coulomb forces were computed with the Particle Mesh Ewald method with a grid spacing of 1 Å. 2 fs timesteps were used with non-bonded interactions calculated every 2 fs and full electrostatics every 4 fs while hydrogens were constrained with the SHAKE algorithm. The cut-off distance was 12 Å with a switching distance of 10 Å and a pair-list distance of 14 Å. Pressure was controlled to 1 atmosphere using the Nosé-Hoover Langevin piston method employing a piston period of 100 fs and a piston decay of 50 fs. Trajectory frames were captured every 100 picoseconds. Ice fractions were determined by measuring the ratio of mobile to immobile water molecules. Water molecules were determined to be immobile, (i.e., ice), if the average oxygen position of water had moved less than 0.8 Å over three successive trajectory frames (i.e., 200 ps). The initial pdb model of the sbwAFP at the ice/water interface is included in the supplementary data (*Supplementary file 1*).

## Calculating critical size of spherical ice embryo at 1 K supercooling

Using *Equation 1* for the case of spherical geometry ($A_g = 2$),

$$R \approx \frac{M_w \sigma}{L \rho_i} \frac{2T_0}{(T_0 - T_r)},$$

where $R$ is the radius, $M_w$ the molecular weight of water (0.018 kg/mol), $L$ the latent heat of melting of water ice ($6.02 \times 10^3$ J/mol) (*Vega et al., 2005*), $\sigma$ is the ice-water surface energy $29.1 \times 10^{-3}$ J/m$^2$ (*Handel et al., 2008*), $\rho_i$ is the density of ice (917 kg/m$^3$), $T_0$ the normal melting temperature (273.15 K), and $T_r$ is the melting temperature depressed by curvature (272.15 K). These values give $R = 518$ Å. A sphere of this radius has a volume of $5.82 \times 10^8$ Å$^3$ and would contain $1.8 \times 10^7$ water molecules.

Using parameters derived from the TIP4P water model (*Vega et al., 2005*; *Handel et al., 2008*), the corresponding values are $T_0$ of 230.5 K, $\sigma$ of $23 \times 10^{-3}$ J/m$^2$, $\rho_i$ of 944 kg/m$^3$ and $L$ of $4.4 \times 10^3$ J/mol. Using the cylindrical version of this Gibbs–Thomson equation ($A_g = 1$) and a freezing point depression of 5 K, we find the critical cylindrical radius to be 46 Å.

## Acknowledgements

This research was supported by a Victorian Life Sciences Computation Initiative (VLSCI) grant number VR0064 on its Peak Computing Facility at the University of Melbourne, an initiative of the Victorian Government, Australia, and by a grant from the Australian Research Council (DP110103388) and computational resources from the Victorian Partnership for Advanced Computing. We thank Dr Christina Hall for discussion and manuscript preparation.

## Additional information

### Funding

| Funder | Grant reference | Author |
| --- | --- | --- |
| Australian Research Council (ARC) | DP110103388 | Sneha E Abraham, Angus Gray-Weale |
| University of Melbourne | Victorian Life Sciences Computation Initiative grant number VR0064 | Michael J Kuiper |
| Victorian Government | | Michael J Kuiper |
| Victorian Partnership for Advanced Computing | | Michael J Kuiper |

The funders had no role in study design, data collection and interpretation, or the decision to submit the work for publication.

### Author contributions

MJK, AG-W, Conception and design, Acquisition of data, Analysis and interpretation of data, Drafting or revising the article, Contributed unpublished essential data or reagents; CJM,

Conception and design, Analysis and interpretation of data, Drafting or revising the article; SEA, Conception and design, Acquisition of data, Analysis and interpretation of data

## Additional files

### Supplementary file

• Supplementary file 1. sbwAFP ice/water interface model. A pdb files of the initial sbwAFP ice water model is included. Constrained water molecules of the ice seed and disordered waters have a 1.0 value in the beta column.

### Major dataset

The following previously published dataset was used:

| Author(s) | Year | Dataset title | Dataset ID and/or URL | Database, license, and accessibility information |
|---|---|---|---|---|
| Leinala EK, Davies PL, Doucet D, Tyshenko MG, Walker VK, Jia Z | 2002 | Choristoneura Fumiferana (Spruce Budworm) Antifreeze Protein Isoform 501 | http://www.rcsb.org/pdb/explore/explore.do?structureId=1m8n | Publicly available at RCSB Protein Data Bank (1m8n). |

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
