## [Decision Letter]

Thank you for sending your work entitled “The biological function of an insect antifreeze protein simulated by molecular dynamics” for consideration at *eLife*. Your article has been favorably evaluated by John Kuriyan (Senior editor), a Reviewing editor, and two reviewers.

The Reviewing editor and the reviewers discussed their comments before we reached this decision, and the Reviewing editor has assembled the following comments to help you prepare a revised submission.

This paper explores the molecular mechanism of an antifreeze protein with the help of molecular dynamics simulations. The simulations show that spruce budworm antifreeze protein (sbwAFP) stalls ice growth at the ice prismatic plane in a manner consistent with the Gibbs-Thomson effect. While the study is clearly interesting, a number of serious criticisms have been raised, in particular concerning novelty (in light of earlier simulation studies of ice-growth inhibition by the same protein), the choice of the water model (with a TIP4P having a very low freezing point), the assumed ice growth mode, and the calculation of binding free energies. The points listed below would have to be convincingly addressed for the paper to be publishable in *eLife*.

Major comments:

1) Novelty. The present study is closely related to the earlier work of Furukawa and Nada. In Nada, 2001, the same protein was studied, and very similar conclusions were reached. In particular, the ice melting point depression due to the Gibbs-Thomson effect has already been reported by Nada and Furukawa (Nada, 2008 and Nada, 2001). Specific questions to address are: What distinguishes the present study from these earlier simulations? Is the claim that “to the best of our knowledge [the Gibbs-Thomson phenomena] has not been observed directly” justified? What are the differences to these earlier studies? Does the broader temperature range, the consideration of mutant proteins, and the longer simulations lead to substantially different conclusions or a deeper understanding? It has to be clear that the present paper constitutes a major scientific advance.

2) Water model. The simulations are performed with the TIP4P model. However, TIP4P gives a only a poor rendering of the water phase diagram. In particular, the ice melting point is far too low. As a result, the simulations were carried out at ∼230 K. However, the diffusion of water and protein at that temperature is very slow, resulting in poor sampling. There are several models in which the reproduction of the phase diagram is much better than in the TIP4P model, e.g., the six-site model and TIP5P/ice model. Could the authors comment about the dependence of the simulation results on the water potential model? It would be reassuring if simulations with a different water model gave similar results.

3) Ice growth mode. The simulations are set up assuming a step growth mode on the ice prismatic plane. Is this realistic? At 1 atm pressure, ice crystals grown from water do not have prismatic plane facets. This means that prismatic planes should not grow by a step-growth mode. Prismatic plane facets appear on grown ice crystal shapes only when pressure is extremely high (M. Maruyama, J Cryst. Growth 275, 2005, 598). The formation of hexagonal column ice crystals in the presence of sbwAFP suggests that sbwAFP binds to the prismatic plane and inhibit ice growth on the plane, but it does not necessarily mean that the prismatic plane growth occurs by a step growth mode. Would the results be affected if a different, more typical growth mechanism were considered?

4) Estimate of binding free energy and enthalpy. The calculated binding free energies are questionable because of several issues: (i) The variance in the calculated work values is very large (several kT), which makes the application of perturbative methods problematic; (ii) Entropic effects (e.g., free rotation of the molecule in the dissociated state) are probably not sampled correctly; (iii) It is not clear how the work is calculated; (iv) It is not clear if the free energy contains any contributions from the harmonic restraint that would have to be subtracted for a proper binding free energy (see, e.g., Hummer, Szabo, Acc. Chem. Res. 2005); (v) The enthalpy estimate in terms of hydrogen bond numbers does not seem to include hydrogen bonds of free water with ice. To get a more controlled estimate of the binding free energy, it would be important to include also the reverse process of binding and use (variants of) the Crooks fluctuation theorem. In its present form, the binding free energy calculations seem to be more confusing than helpful. They may be more suitable for a separate technical publication, in which details can be explored more carefully.

[Editors' note: further revisions were requested prior to acceptance, as described below.]

Thank you for resubmitting your work entitled “The biological function of an insect antifreeze protein simulated by molecular dynamics” for further consideration at *eLife*. Your revised article has been evaluated by John Kuriyan (Senior editor) and a member of the Board of Reviewing Editors, and the two reviewers who read your original submission. The manuscript has been improved but there are major remaining issues that need to be addressed before acceptance, as outlined below in the summary statement and review prepared by the Reviewing Editor.

It should be possible for you to address the issues raised by this round of review by editing your manuscript in response to the issues raised, without the need for any further calculations. Please pay particular attention to the fact that the Reviewing Editor asks for the elimination of the data and discussion concerning the calculation of ice-binding free energies. Our proceeding further with a revised manuscript is contingent on this being done, because the editor and the reviewers have identified gaps in this analysis.

Summary statement from the Reviewing Editor:

Molecular dynamics simulations are used to study the trapping of an advancing ice-water interface by antifreeze proteins. The simulations are insightful and interesting, and shed light on an important but poorly understood process at the interface of biology and physics. In their rebuttal and revision, the authors have made a good effort to address the concerns raised in the previous reviews. However, there are important issues that have not been adequately addressed.

To make the paper acceptable for publication in *eLife* will require (1) a considerably more nuanced discussion of preceding work, and (2) the elimination of the data and discussion of the calculation of ice-binding free energies. Detailed reasons are given below. In addition, (3) the discussion of the semi-quantitative agreement with the Gibbs-Thomson formalism, as given in [Disp-formula equ1] should better reflect evident uncertainties in this comparison. Also, (4) a clearer justification of the choice of ice-growth model should be included in the paper, not just in the rebuttal. In the following, I discuss these four points in detail.

First, I find the discussion of earlier studies inadequate. I am particularly troubled by statements such as “previous AFP simulations have fallen short of complete, convincing molecular descriptions of an AFP-ice interaction” and “recent advances in computational science now make complete simulations feasible and we report them here.” What do the authors mean by their simulations being “complete”? Is there nothing to be done in this field after this study? This is certainly not the case, with many questions remaining open, e.g., concerning the water model with its freezing point of ∼240 K, the quantification of the phenomenological theory in [Disp-formula equ1], the mode of ice growth, or the calculation of a binding free energy. A fairer discussion of earlier results and a more modest presentation of the new results are in place. Such a discussion should explicitly mention the work of Nada and Furukawa on the use of MD simulations to study ice growth kinetics around AFPs, as influenced by the Gibbs-Thomson effect.

Second, there are major issues with the calculation of the protein-ice binding free energy. Quoting Reviewer #2, in revising their manuscript, the authors have substantially increased the details of their simulation methodology and have now included a much-expanded discussion on the calculation of the binding free energy. Unfortunately these details now render these calculations and the conclusions derivable from them somewhat questionable. These calculations could be presented separately in a more technical journal to do justice to the many challenges involved in such demanding calculations.

The instantaneous switching of the control parameters (either the spring constant or harmonic minima) is expected to yield reliable free energy estimates only when such changes result in dissipative work values on the order of kT (where the exponential of the work is well approximated by perturbation theory). The accumulated work values in Figure 9 place this study far outside of this regime, and it is likely the poor sampling of exponentially rare work values that lead to systematic errors and ultimately the disagreement between various protocol estimates of the binding energy discussed in the appendix. Contrary to the authors' claims in their revision, simulating both forward and backward protocols to converge the binding energy would be hugely beneficial precisely because of the irreversibility of binding that the authors allude to. Only in instances of microscopically reversible dynamics do the two procedures yield redundant information. The authors may, e.g., consider Dellago and Hummer, 2013, for guidelines concerning the application of these equations (Dellago, C. and Hummer, G., “Computing Equilibrium Free Energies Using Non-Equilibrium Molecular Dynamics”, Entropy 16.1 (2013): 41-61).

The discussion of the binding free energy in terms of the number of broken hydrogen bonds is also too vague to be useful. As pointed out by Reviewer #2, the number listed for the hydrogen bond enthalpy seems to be judiciously picked to yield agreement with the calculated binding energy (the reference pointed to in the manuscript lists a slightly different value) and moreover hydrogen bonding energies are known to be rapidly varying functions of bonding geometry. The authors' recent correction of this estimate to reflect an activation energy rather than the binding energy presupposes that the binding is barrierless for the equivalence of these two quantities to be thermodynamically meaningful and that it is necessary to break hydrogen bonds along the reaction coordinate for this estimate to be kinetically meaningful. From the data provided and cited, it is not clear that either of those assumptions is valid.

I want to add to the comments of Reviewer #2 that even the definition of the binding free energy is unclear. Binding equilibria from a bulk phase to a surface have dimensional contributions that require a choice of standard state. Furthermore, does the calculated free energy, fully or partially, include the entropic contribution from a substantial increase in rotational freedom? Probably not, since the time to sample full rotation in the dissociated state has been too small. On the technical side, I wonder how the work was calculated (i.e., which formula was used and implemented), in particular for the cases in which the spring constant was changed with time.

Third, Reviewer #2 points out that the near quantitative agreement of the simulation results with the radius of curvature calculated from [Disp-formula equ1] seems fortuitous. Is there a reason to believe that the surface tension or enthalpy of fusion for the water model used are the same as in experiment? Can experimental parameters taken from a website (http://www.its.caltech.edu/∼atomic/snowcrystals/ice/ice.htm) be used to explain the observations of the simulations? Such a validation procedure seems questionable. There are also technical issues with the application of [Disp-formula equ1] to a molecular simulation system. As discussed previously (Findenegg et al., “Freezing and melting of water confined in silica nanopores”, ChemPhysChem 9.18 (2008): 2651-2659), how exactly the interface and the radius of curvature are calculated is not unique. The implications on the calculated value of the radius would need to be discussed for this quantitative comparison to be meaningful.

Finally, Reviewer #1 asks for further clarification in the paper concerning the ice-growth mode. In the rebuttal, the authors mention: “Our preliminary work using ‘perfect’ crystals did not grow well, extending the ice front very slowly if at all. We realised this was because one had to effectively wait for a 2 dimensional nucleation event on the prism plane to occur and reach a critical size before the next layer of ice would grow.” The authors should include this explanation in the paper to justify their assumption of a step-growth mode on the prismatic plane.

---

## [Author Response]

*1) Novelty. The present study is closely related to the earlier work of Furukawa and Nada. In Nada, 2001, the same protein was studied, and very similar conclusions were reached. In particular, the ice melting point depression due to the Gibbs-Thomson effect has already been reported by Nada and Furukawa (Nada*, *2008 and Nada, 2001). Specific questions to address are: What distinguishes the present study from these earlier simulations?*

Our studies are the first to show reproducible ice binding of an AFP as it migrates freely from the aqueous phase to the ice surface and the first to show total ice inhibition consistent with the Gibbs-Thomson mechanism while also revealing surprising details of the water arrangements at the interface layer. It is also one of the first, by including mutant control proteins, to demonstrate how crucial the local water binding arrangement is for AFP activity.

Nada and Furukawa performed superficially similar simulations in 2011, however their study has serious shortcomings. Firstly, their two AFPs initial conformations was pre-determined using a docking routine on an idealized ice prism surface without liquid water. As their simulations were also quite short, (only 2 simulations of 6 nanoseconds), the pre-positioning of AFP at the ice interface compounds the biases by limiting the sampling of interfacial water.

Secondly, Nada and Furukawa have not clearly demonstrated total ice inhibition, but only a slowing of the ice growth rate (shown in their Figure 5). Their ice growth never plateaued and their simulations are too short to allow proper sampling or protein diffusion at the ice interface. It was perhaps premature on their part to attribute the curvature to the Gibbs-Thomson effect; indeed, one could argue that over their simulation time-scale a similar sized non-AFP protein may have caused an equivalent ice-distortion, but this was not tested with a control.

Nada and Furukawa based their conclusions on limited sampling of 2 simulations of only 6 nanoseconds length. Our study has the fortune to be based on sampling of a long 3000 nanosecond simulation plus 32 wildtype binding simulations (each of 150 to 250 nanoseconds in length) and 16 simulations of the mutant form (each of 250 nanoseconds in length) totalling over 12 microseconds for the binding/inhibition part of our experiments alone.

We have modified our text to emphasise these advances over earlier studies (Introduction):

“Impeded by computational constraints, previous AFP simulations have fallen short of complete, convincing molecular description of an AFP-ice interaction. […] Recent advances in computational science now make complete simulations feasible and we report them here.”

Is the claim that “to the best of our knowledge [the Gibbs-Thomson phenomena] has not been observed directly” justified?

Upon wider reading, we did find references that refer to observing the Gibbs-Thomson effect in nanomaterials research dealing with semiconductors. Consequently, strictly speaking the original wording would be invalid. We further consulted with an expert in the field, Dr. Charlie Knight, who agreed that in context of AFPs, resolving the ice/water interface at the scale of a few tens of nanometers is nearly impossible, even with atomic force microscopy.

We have thus modified the subject text to be more specific (subsection titled “Adsorption inhibition”):

“Though the Gibbs-Thomson effect is widely accepted as the basis of AFP activity, resolving these nanoscale features at the ice/water interface has so far not been attained experimentally.”

*2) Water model. The simulations are performed with the TIP4P model. However, TIP4P gives a only a poor rendering of the water phase diagram. In particular, the ice melting point is far too low. As a result, the simulations were carried out at ∼230 K. However, the diffusion of water and protein at that temperature is very slow, resulting in poor sampling. There are several models in which the reproduction of the phase diagram is much better than in the TIP4P model, e.g., the six-site model and TIP5P/ice model. Could the authors comment about the dependence of the simulation results on the water potential model? It would be reassuring if simulations with a different water model gave similar results*.

The TIP4P water model does indeed have a well-known discrepancy with regards to its freezing temperature. We would have preferred to use a more sophisticated water model such as the TIP5P/ice, however the only water models that were supported by the NAMD software we used were TIP3P and TIP4P. Other MD programs such as Amber can support more water models, but at significantly slower performance. NAMD was the only software we had that would scale efficiently on our BlueGene clusters to get the required timescales. Also the TIP4P model has been used in a number of notable ice simulation papers, namely Mochizuki et al., (Nature 498.7454 (2013): 350-354) and Matsumoto et al., (Nature 416.6879 (2002): 409-413).

Also, we do agree the diffusion of water is slow at ∼230K but with our longer simulations we have seen the fluctuations in the ice front; observing the front grow and retreat. The fact that we show the molecules are able to reversibly arrange themselves into a crystal structure shows sufficient sampling the relevant regions of phase space. It is certainly a good point to be wary of, so we have added an additional note in the text (in the subsection “Adsorption inhibition”):

“The extent of the inhibited ice front fluctuated, advancing and retreating in the range of 5 to 8 Å showing the reversible arrangement of water molecules to the crystal ice lattice, (this is an important indication of reaching equilibration and adequate sampling given the reduced water diffusion at these low temperatures).”

Overall, we are confident that the TIP4P model can be used to give good representative behaviour of AFP activity, as long as sufficient sampling is done to compensate for slower water diffusion.

3) Ice growth mode. The simulations are set up assuming a step growth mode on the ice prismatic plane. Is this realistic?

Ice crystals are typically riddled with imperfections as shown by various vacuum etching studies, which serve as growth mechanisms such as screw dislocations (Sinha, N. K. (1977). Dislocations in ice as revealed by etching. Philosophical Magazine, 36(6), 1385-1404).

Our preliminary work using “perfect” crystals did not grow well, extending the ice front very slowly if at all. We realised this was because one had to effectively wait for a 2 dimensional nucleation event on the prism plane to occur and reach a critical size before the next layer of ice would grow. The tilt of the seed crystal and resultant step growth ensures a constant ice growth mechanism with similar effect to a screw dislocation. We believe this to be not only a realistic model of ice growth, but immensely practical to introduce as it greatly reduces the computational time necessary to achieve a certain rate of growth.

*At 1 atm pressure, ice crystals grown from water do not have prismatic plane facets. This means that prismatic planes should not grow by a step-growth mode. Prismatic plane facets appear on grown ice crystal shapes only when pressure is extremely high (M. Maruyama, J Cryst. Growth 275, 2005, 598)*.

Our model ice face is only centred on the prism ice face for practical purposes. It would be incorrect to take this as representative of a prism plane of macroscopic ice crystal.

The formation of hexagonal column ice crystals in the presence of sbwAFP suggests that sbwAFP binds to the prismatic plane and inhibit ice growth on the plane, but it does not necessarily mean that the prismatic plane growth occurs by a step growth mode. Would the results be affected if a different, more typical growth mechanism were considered?

The step growth model we have developed here is a convenient way to test the AFP activity in the presence of a growing ice front. With a much larger system, (and much more computational overhead!), one could create other crystal growth initiators conditions, such as a screw dislocation for similar layer-by-layer growth mechanism. We would expect similar results, as steps originating from screw dislocations are still essentially step growth. It would certainly be interesting to test AFP activity around such a dislocation in simulation, but that would currently require yet another massive increase in computational capability.

*4) Estimate of binding free energy and enthalpy. The calculated binding free energies are questionable because of several issues: (i) The variance in the calculated work values is very large (several kT), which makes the application of perturbative methods problematic*.

If the error bounds seem large, it is because we have been most careful and conservative in our estimation of the free energy. The estimates are as derived and analysed by Gore et al*.*. We are aware of no superior method of error estimation. We do not see that larger error and bias estimates raise the possibility of sources of error not captured by the estimates used, though we have carefully examined the literature on the use of the Jarzynski theorem.

Further to these error estimates, we have examined the effect of varying the speed of the protein, and checked that calculation of ΔG for the motion of the protein through pure water is appropriately close to 0. We used two independent sets of trajectories and identified their strengths and weaknesses. Our paper depends only on the demonstration that the binding is irreversible. We submit that this is clearly shown by our results. We have now also expanded and clarified the discussion of the Jarzynski method.

*(ii) Entropic effects (e.g., free rotation of the molecule in the dissociated state) are probably not sampled correctly*.

In the dissociated state, the protein is surrounded by water on all sides, to a distance of at least the correlation length in pure water. This state on average is symmetrical, and so sampling more orientations would not alter the thermodynamics of this state. More broadly, and not only for the purposes of the Jarzynski calculation, equilibration is important. As discussed in the main paper, we see stabilisation of the ice front and protein in each structure examined.

*(iii) It is not clear how the work is calculated*.

We have greatly expanded and clarified the report of this calculation. There are too many changes for these to be set out in full here, but for example the relevant Methods section now opens with an outline of its components, and how they fit together.

Other salient improvements are in the Methods section, where the role of the error and bias estimates is discussed, as well as the use of tests of the bias method.

*(iv) It is not clear if the free energy contains any contributions from the harmonic restraint that would have to be subtracted for a proper binding free energy (see, e.g., Hummer, Szabo, Acc. Chem. Res. 2005)*.

Fortunately the free energy of a harmonic oscillator is exactly and easily calculable, and the force constants we use correspond to free energies smaller than the random errors.

The work by Hummer and Szabo pertains to the application of Jarzynski's theorem to single molecule force spectroscopy, rather than to a simulation. The important correction that they consider concerns the measurement of force as a function of time, when the work W in Jarzynski's theorem is the integral of force with respect to distance. This is a subtle issue, but one that we circumvent in a simulation because the exact value of the work W is available for each trajectory.

*(v) The enthalpy estimate in terms of hydrogen bond numbers does not seem to include hydrogen bonds of free water with ice*.

Estimating the hydrogen bonds of free water with ice is problematic as it is difficult to define which molecule is water and which is ice. We have clarified our original estimate and argument to consider the initial step of removing the protein from the ice as requiring simultaneously breaking the 12 hydrogen bonds we observe made from the ordered water to the ice and relating this to the activation free energy for the removal of the protein from the ice.

To get a more controlled estimate of the binding free energy, it would be important to include also the reverse process of binding and use (variants of) the Crooks fluctuation theorem.

The individual trajectories are done deterministically and so doing more trajectories is the same as studying also the reverse process. To use the Crooks theorem on top of Jarzynski would be redundant.

*In its present form, the binding free energy calculations seem to be more confusing than helpful. They may be more suitable for a separate technical publication, in which details can be explored more carefully*.

We have made improvements on the clarity of the binding free energy calculations as detailed above.

[Editors' note: further revisions were requested prior to acceptance, as described below.]

*To make the paper acceptable for publication in* eLife *will require (1) a considerably more nuanced discussion of preceding work, and (2) the elimination of the data and discussion of the calculation of ice-binding free energies. Detailed reasons are given below. In addition, (3) the discussion of the semi-quantitative agreement with the Gibbs-Thomson formalism, as given in*
[Disp-formula equ1]
*should better reflect evident uncertainties in this comparison. Also, (4) a clearer justification of the choice of ice-growth model should be included in the paper, not just in the rebuttal. In the following, I discuss these four points in detail.*

*First, I find the discussion of earlier studies inadequate. I am particularly troubled by statements such as “previous AFP simulations have fallen short of complete, convincing molecular descriptions of an AFP-ice interaction” and “recent advances in computational science now make complete simulations feasible and we report them here.” What do the authors mean by their simulations being “complete”? Is there nothing to be done in this field after this study? This is certainly not the case, with many questions remaining open, e.g., concerning the water model with its freezing point of ∼240 K, the quantification of the phenomenological theory in*
[Disp-formula equ1]*, the mode of ice growth, or the calculation of a binding free energy. A fairer discussion of earlier results and a more modest presentation of the new results are in place. Such a discussion should explicitly mention the work of Nada and Furukawa on the use of MD simulations to study ice growth kinetics around AFPs, as influenced by the Gibbs-Thomson effect*.

We have greatly elaborated on prior work as suggested by the reviewers, including citing an additional 8 references and adding three paragraphs in relation to the historical context of computational simulation of AFPs, with particular mention of the work from Nada and Furukawa.

*Second, there are major issues with the calculation of the protein-ice binding free energy. Quoting Reviewer #2, in revising their manuscript, the authors have substantially increased the details of their simulation methodology and have now included a much-expanded discussion on the calculation of the binding free energy. Unfortunately these details now render these calculations and the conclusions derivable from them somewhat questionable. These calculations could be presented separately in a more technical journal to do justice to the many challenges involved in such demanding calculations*.

*The instantaneous switching of the control parameters (either the spring constant or harmonic minima) is expected to yield reliable free energy estimates only when such changes result in dissipative work values on the order of kT (where the exponential of the work is well approximated by perturbation theory). The accumulated work values in Figure 9 place this study far outside of this regime, and it is likely the poor sampling of exponentially rare work values that lead to systematic errors and ultimately the disagreement between various protocol estimates of the binding energy discussed in the appendix. Contrary to the authors' claims in their revision, simulating both forward and backward protocols to converge the binding energy would be hugely beneficial precisely because of the irreversibility of binding that the authors allude to. Only in instances of microscopically reversible dynamics do the two procedures yield redundant information. The authors may, e.g., consider Dellago and Hummer, 2013, for guidelines concerning the application of these equations (Dellago, C. and Hummer, G., “Computing Equilibrium Free Energies Using Non-Equilibrium Molecular Dynamics”, Entropy 16.1 (2013): 41-61)*.

*The discussion of the binding free energy in terms of the number of broken hydrogen bonds is also too vague to be useful. As pointed out by Reviewer #2, the number listed for the hydrogen bond enthalpy seems to be judiciously picked to yield agreement with the calculated binding energy (the reference pointed to in the manuscript lists a slightly different value) and moreover hydrogen bonding energies are known to be rapidly varying functions of bonding geometry. The authors' recent correction of this estimate to reflect an activation energy rather than the binding energy presupposes that the binding is barrierless for the equivalence of these two quantities to be thermodynamically meaningful and that it is necessary to break hydrogen bonds along the reaction coordinate for this estimate to be kinetically meaningful. From the data provided and cited, it is not clear that either of those assumptions is valid*.

*I want to add to the comments of Reviewer #2 that even the definition of the binding free energy is unclear. Binding equilibria from a bulk phase to a surface have dimensional contributions that require a choice of standard state. Furthermore, does the calculated free energy, fully or partially, include the entropic contribution from a substantial increase in rotational freedom? Probably not, since the time to sample full rotation in the dissociated state has been too small. On the technical side, I wonder how the work was calculated (i.e., which formula was used and implemented), in particular for the cases in which the spring constant was changed with time*.

We have removed all results and discussions regarding the free-energy calculations. We agree that this aspect of the work is difficult and perhaps better suited in a more technical journal where we can explore technical and theoretical details fully.

We greatly appreciate the detailed critique of the free-energy methods we originally employed. Since we have removed the controversial components raised by the reviewers, specifically instantaneous switching of control parameters and binding energy in terms of broken hydrogen bonds from this manuscript, we shall address them and binding energy definitions in subsequent work with and not elaborate them here.

*Third, Reviewer #2 points out that the near quantitative agreement of the simulation results with the radius of curvature calculated from*
[Disp-formula equ1]
*seems fortuitous. Is there a reason to believe that the surface tension or enthalpy of fusion for the water model used are the same as in experiment? Can experimental parameters taken from a website (**http://www.its.caltech.edu/∼atomic/snowcrystals/ice/ice.htm**) be used to explain the observations of the simulations? Such a validation procedure seems questionable. There are also technical issues with the application of*
[Disp-formula equ1]
*to a molecular simulation system.* A*s discussed previously (Findenegg et al., “Freezing and melting of water confined in silica nanopores”, ChemPhysChem 9.18 (2008): 2651-2659), how exactly the interface and the radius of curvature are calculated is not unique. The implications on the calculated value of the radius would need to be discussed for this quantitative comparison to be meaningful*.

We have recalculated the radius of curvature of ice using parameters derived from the TIP4P model from literature values of Vega et al., and Handel et al*.,* leading to a smaller radius value of 46 Å compared to our original 59 Å figure. We agree the quantitative comparison is problematic and have dropped the direct reference in favour for a qualitative comparison and a brief mention of the difficulties in measuring curvature from a simulation. We have replaced the shaded 60 Å radius circular segment in Figure 8, with that of a 46 Å radius segment equal to that of the theoretical radius of curvature value derived from TIP4P model parameters.

*Finally, Reviewer #1 asks for further clarification in the paper concerning the ice-growth mode. In the rebuttal, the authors mention: “Our preliminary work using ‘perfect’ crystals did not grow well, extending the ice front very slowly if at all. We realised this was because one had to effectively wait for a 2 dimensional nucleation event on the prism plane to occur and reach a critical size before the next layer of ice would grow.” The authors should include this explanation in the paper to justify their assumption of a step-growth mode on the prismatic plane*.

We have further clarified the step growth mode of the ice model by mentioning our earlier experience of slow growth of ‘perfect’ crystals and referencing work of Frank and Hobbs in relation to step propagation as a mechanism crystal growth.